# Machine Learning Algorithms for Predicting Stunting among Under-Five Children in Papua New Guinea

**DOI:** 10.3390/children10101638

**Published:** 2023-09-30

**Authors:** Hao Shen, Hang Zhao, Yi Jiang

**Affiliations:** School of Public Health, Chongqing Medical University, Chongqing 400016, China; 2021110664@stu.cqmu.edu.cn (H.S.); 2021110669@stu.cqmu.edu.cn (H.Z.)

**Keywords:** stunting, children, machine learning, Papua New Guinea

## Abstract

Preventing stunting is particularly important for healthy development across the life course. In Papua New Guinea (PNG), the prevalence of stunting in children under five years old has consistently not improved. Therefore, the primary objective of this study was to employ multiple machine learning algorithms to identify the most effective model and key predictors for stunting prediction in children in PNG. The study used data from the 2016–2018 Papua New Guinea Demographic Health Survey, including from 3380 children with complete height-for-age data. The least absolute shrinkage and selection operator (LASSO) and random-forest-recursive feature elimination were used for feature selection. Logistic regression, a conditional decision tree, a support vector machine with a radial basis function kernel, and an extreme gradient boosting machine (XGBoost) were employed to construct the prediction model. The performance of the final model was evaluated using accuracy, precision, recall, F1 score, and area under the curve (AUC). The results of the study showed that LASSO-XGBoost has the best performance for predicting stunting in PNG (AUC: 0.765; 95% CI: 0.714–0.819) with accuracy, precision, recall, and F1 scores of 0.728, 0.715, 0.628, and 0.669, respectively. Combined with the SHAP value method, the optimal prediction model identified living in the Highlands Region, the age of the child, being in the richest family, and having a larger or smaller birth size as the top five important characteristics for predicting stunting. Based on the model, the findings support the necessity of preventing stunting early in life. Emphasizing the nutritional status of vulnerable maternal and child populations in PNG is recommended to promote maternal and child health and overall well-being.

## 1. Introduction

Stunting has been defined as the lack of height relative to age in children [1] and is the most prevalent form of child malnutrition [2]. Stunting occurs mainly during the critical window of 0–24 months [3], which is the most sensitive period of child growth and development. Stunting was found to be especially vulnerable to environmentally modifiable factors [4]. This growth deficit continues to accumulate and worsens during early childhood (0–5 years) due to continued exposure to adverse environmental factors such as feeding, infections, and psychosocial factors [5].

The consequences of stunting observed within the first five years of life are far-reaching, encompassing increased morbidity and mortality, impaired cognitive development, poorer academic performance, physical developmental deficits, and diminished economic productivity [6]. Despite some studies suggesting the possibility of catch-up growth in stunted children, there is no conclusive evidence to support the full reversal of the early-life effects of stunting [7,8].

As of 2020, approximately 149 million children under the age of five remain affected by stunting worldwide with the overwhelming majority of cases (96.7%) occurring in low- and middle-income countries [9]. It is evident that stunting in children poses a significant global health challenge [1]. In response, Target 2.2 of the Sustainable Development Goals (SDGs) states that all forms of malnutrition should be eliminated by 2030, which includes stunting in children under five years of age [10].

Despite impressive achievements in reducing stunting in the Western Pacific Region, progress remains slow in some countries [11]. Papua New Guinea (PNG) is among the countries where stunting rates among children under five years old have persistently failed to improve, rising from 47.2% in 2000 to 48.4% in 2020. Surprisingly, this trend contradicts that of PNG’s rapid economic growth [12]. The increase in resources and wealth has not improved the nutritional status of children [13]. Consequently, there is a need to address stunting in children under five years of age in PNG as a serious public health issue.

Previous studies from PNG have explored factors associated with stunting, such as regional disparities, wealth indices, maternal education level, and childhood vaccinations [14,15,16,17]. However, these studies often relied on limited data and lacked national representativeness, limiting the generalizability of their results to the wider PNG population. A few studies have applied nationally representative data from the 2009–2010 Papua New Guinea Household Income and Expenditure Survey (PNG HIES) [18,19] to examine stunting prevalence variations across different regions in PNG. However, the timeliness of the data restricted their scope, and they only adjusted for a limited number of confounding variables.

Machine learning (ML) has emerged as a powerful data-mining technique that is particularly adept at handling high-dimensional and nonlinear relationships [20,21], surpassing classical statistical models in many aspects. As a result, ML algorithms have found widespread application in the exploration of the social determinants of health (SDHs) [21]. The application of algorithms such as decision trees (DTrees), random forests (RFs), support vector machines (SVMs), gradient boosting machines (GBMs), extreme gradient boosting machines (XGBoosts), and neural networks (ANNs) is commonly used in studies exploring the factors associated with stunting in children [22,23,24,25,26,27,28]. Evidence from Ethiopia, Tanzania, and Bangladesh [23,26,27,28] showed that traditional logistic regression (LR) often fails to achieve optimal performance in predicting stunting in children compared to other ML algorithms. Consequently, the application of multiple ML algorithms becomes imperative to identify the best predictive model.

Feature selection (FS), a technique aimed at reducing dimensionality, plays a crucial role in optimizing an algorithm’s predictive performance by eliminating redundant, irrelevant, and noisy features [29]. FS is usually categorized into filtered, embedded, wrapper, and hybrid methods [30]. Embedded methods employ built-in feature selection methods to optimize objective functions or classifiers [31], such as decision trees and the least absolute shrinkage and selection operator (LASSO). Conversely, wrapper methods employ repetitive learning steps and resampling techniques to evaluate feature usefulness and result in enhanced predictive capabilities but at a higher computational cost [32].

Given that the prevalence of stunting in children under five years of age in PNG is still not promising, there is a need for targeted programs and effective interventions. Therefore, the main objective of this study is to apply FS techniques with ML algorithms to train, evaluate, and select the best model for predicting stunting in children under five years of age in PNG based on the nationally representative 2016–2018 Papua New Guinea Demographic Health Survey database (2016–2018 PNG DHS) in addition to obtaining the most important features for predicting stunting. The study’s findings will provide evidence for PNG policy makers to plan scientifically sound programs with integrated interventions to prevent child stunting and protect the health of the most vulnerable subgroups of children. This will help accelerate PNG’s progress in the SDGs related to children’s health.

## 2. Materials and Methods

### 2.1. Data Source

The cross-sectional data used in this study were obtained from the 2016–2018 PNG DHS, which was conducted by the PNG National Statistics Office (NSO). This comprehensive national survey covered individuals aged 15–49 years in PNG with the aim to provide current information on key demographic and health indicators. The survey employed a two-stage stratified sampling method to select approximately 19,200 households, with 18,175 women aged 15–49 from the surveyed households eligible for individual interviews. A total of 15,198 women completed the interviews with a response rate of 84%. Child information was collected from mothers or primary caregivers. Structured questionnaires were applied for data collection, and details about the survey can be found in the 2016–2018 PNG DHS final report [33]. For households where male participants were selected for interviews, the 2016–2018 PNG DHS conducted height, length, weight, and mid-upper arm circumference (MUAC) measurements for eligible children under five years of age using equipment provided by UNICEF [33]. Ultimately, all children under five years of age with complete and valid height-for-age data were included in this study with a total of 3380 children meeting the inclusion criteria.

The 2016–2018 Papua New Guinea Demographic and Health Survey (PNG DHS) received ethical approval from the Institutional Review Board of Inner City Fund International. Additionally, informed consent was obtained from respondents for all interviews. On 17 August 2023, the DHS program approved the use of this dataset for this study. All data were desensitized (anonymized by removing all personal identifiers) before being received by the authors. This study was conducted in accordance with relevant guidelines and regulations regarding the published use of DHS datasets and did not require additional ethical review documentation or informed consent due to the use of open secondary data. Further information about DHS data and ethical standards is available at https://dhsprogram.com/methodology/Protecting-the-Privacy-of-DHS-Survey-Respondents.cfm (accessed on 17 July 2023).

### 2.2. Outcome Variable and Potential Risk Factors

Our outcome variable of interest was stunting, which was coded as a binary variable. According to criteria developed by the WHO in 2006, children with height-for-age z-scores (HAZs) that are 2 standard deviations below the WHO growth standards are recognized as stunted [1] and coded as 1, while all others are coded as 0. The conceptual framework proposed by the United Nations Children’s Nutrition Foundation (UNICEF) illustrates that stunting is attributed to complex contextual, underlying, and direct causes [34]. Therefore, based on the results of previous studies, this study incorporated potential factors and divided them into four main categories: individual characteristics, maternal characteristics, family characteristics, and community characteristics.

Individual characteristics included the child’s gender, age, birth size, birth order, duration of breastfeeding, early breastfeeding, and occurrence of diarrhea and fever in the last two weeks. Maternal characteristics included the mother’s age (years), employment status, occupation, marital status, education level, age at first birth (years), exposure to mass media, and their partner’s age (years), employment status, and education level. Following WHO recommendations [35], early breastfeeding was defined as the initiation of breastfeeding within one hour of delivery. Breastfeeding duration was categorized as never breastfeeding, a breastfeeding duration < 6 months, and a breastfeeding duration ≥ 6 months [36]. Exposure to mass media was based on the frequency of women reading newspapers, watching television, and listening to the radio; access to at least one of these media was considered exposure to mass media [37].

The household characteristics encompassed the sex of the househead, the number of children under five years of age, the number of household members, the type of latrine, the source of drinking water, the type of fuel for the kitchen, and the distance to the health facility. Community characteristics covered the place of residence as well as the region. Based on the WHO/UNICEF guidelines [38], the source of drinking water was categorized as unimproved or improved, and the type of latrine was categorized as unfurnished, unimproved, or improved. Based on WHO indoor air quality guidelines [39], kitchen fuel types were categorized as clean or polluting fuels, where clean fuels included electricity and liquefied petroleum gas. Household wealth was a composite index constructed by the 2016–2018 PNG DHS, where a principal component analysis was applied based on the household’s consumer goods and housing characteristics, forming the corresponding household wealth quintiles: poorest, poorer, middle, richer, and richest [33].

### 2.3. Analytic Strategy

#### 2.3.1. Preprocessing

Data preprocessing was performed using STATA 17.0 statistical software. We conducted an initial screening of categorical and continuous variables using the χ2 (bivariate) test with the Wilcoxon rank sum test, and variables with a *p* > 0.05 were excluded. Descriptive analyses were performed in the form of frequencies for categorical variables and means for continuous variables.

To prepare categorical features for machine learning input, the classical one-hot coding method was employed. After the initial screening, multicategorical variables were converted into multiple binary feature vectors using one-hot coding. This approach ensured that the algorithm did not make erroneous assumptions about variable relationships. Furthermore, the missing indicator method (MIM) was utilized to add indicator metrics to categorical variables containing missing values. The brief analysis steps of this study are shown in Figure 1.

#### 2.3.2. Feature Selection

All subsequent analyses were conducted using RStudio 4.2.3 statistical software. The samples were randomly split into test and training sets at a ratio of 1:9, and feature selection was performed only in the training sets to prevent leakage of test data. To mitigate the risk of overfitting, the AUC value under 10-fold cross-validation (CV) served as the performance evaluation metric [40].

We selected the embedded LASSO and the wrapped random-forest-recursive feature elimination (RF-RFE) for feature selection. Among them, the LASSO controlled model shrinkage via the penalty parameter (λ). By selecting the λ value that produced the highest AUC value, we identified the features with non-zero regression coefficients, forming the optimal feature subset. Alternatively, the RF-RFE measured feature importance using the Gini-coefficient-based Mean Impurity Reduction (MDI) after fitting the RF model. The process was repeated to recursively eliminate irrelevant features until the combination of features with the highest model AUC value was derived. For this analysis, the ntree was set to 500, and the mtry was set to the recommended $\sqrt{p}$ [41].

#### 2.3.3. Machine Learning Algorithms and Hyperparameter Tuning

Grid search (GS) is a traversal search for predefined hyperparameter values performed by a given algorithm. While it is suitable for low-dimensional hyperparameter tuning, it incurs high computational costs [42]. Bayesian optimization based on Gaussian process regression (BO-GPR) leverages a priori information from Gaussian process regression to rapidly converge to the global optimal solution, making it more adept at handling high-dimensional hyperparameter optimization problems with limited iterations [43].

In this study, AUC values were used as performance evaluation metrics for hyperparameter combinations. The GS and BO-GPR strategies under 10-fold cross-validation were applied for the hyperparameter tuning of the following models. In addition, the logistic regression (LR) model, fitted with the generalized linear function (GLM) as a binomial family, required no hyperparameter tuning due to its inherent simplicity and well-defined structure.

The conditional inference tree (CTree) is a special kind of decision tree [44] which embeds a tree regression model into a well-defined conditional inference process. Therefore, easily interpretable classification results can be produced [45]. The CTree included two hyperparameters for controlling the size of the tree’s growth, namely the 1-*p* value (mincriterion) and the maximum depth of the tree (maxdepth) [46]. In selecting the predefined hyperparameter values, caution was exercised, and insights from the relevant literature [47,48,49] were taken into consideration. GS was applied to search for the optimal values of maxdepth and mincriterion, which were confined to the following range:(1)maxdepth=[1,30]
(2)$$mincriterion=\left\{0.900,0.950,0.990\right\}$$

The support vector machine (SVM) is a versatile algorithm used for addressing classification and regression problems. It possesses the capability to linearly classify data while also employing kernel tricks to handle nonlinear data challenges [50]. For instance, the radial basis function (RBF) kernel transforms the input space into a high-dimensional feature space, facilitating the modeling of nonlinear data [51]. In this study, an SVM with a radial basis function kernel with fewer hyperparameters was used to categorize the data. The hyperparameters requiring tuning were the penalty function (*C*) and kernel parameters (*σ*). Following the recommendations of related studies [52,53,54], we applied GS to search for the best *C* and *σ* in the following predefined set:(3)$$C=2^{\left\{-5,-3,-1,1,3,5,7,9,11,13,15\right\}}$$
(4)$$\sigma =2^{\left\{-15,-13,-11,-9,-7,-5,-3,-1,1,3,5,7,9,11\right\}}$$

The extreme gradient boosting machine (XGBoost) is an extensible and integrated algorithm based on gradient boosting decision trees, which is known for its exceptional ability to push the computational power of boosting trees to new limits [55]. The performance of XGBoost was highly dependent on optimizing a large number of hyperparameters, which are summarized as follows: the maximum number of boosting iterations (nrounds), the learning rate (eta), the minimum loss reduction (gamma), the minimum weight sum of instances (min child weight), the maximum depth (maxdepth), the subsample percentage (subsample), and the column-sample-by-tree ratio of subsamples (colsample bytree). Following the recommendations of related studies [56,57,58], we employed BO-GPR to search for the hyperparameters of XGBoost in the ranges presented in Table 1.

#### 2.3.4. Model Performance Evaluation

The final performance of the model in the test set was measured with the AUC, accuracy, precision, recall, and F1 score. Acknowledged as the main performance metric, the AUC gave the overall model performance at each possible classification threshold. The confusion matrix was a square matrix including the True Positive (*TP*), True Negative (*TN*), False Positive (*FP*), and False Negative (*PN*), allowing the extraction of the above-mentioned one-dimensional performance metrics from it [59].

*Accuracy*, defined as the ratio of the number of correct predictions to the total number of predictions, was the most common measure of overall prediction performance.
(5)$$Accuracy=\frac{TP+TN}{TP+TN+FP+PN}$$

*Precision* was defined as the ratio of the number of correct positive predictions to the total number of positive predictions, reflecting the consistency of the predictions with the positive cases in the test set.
(6)$$Precision=\frac{TP}{TP+FP}$$

*Recall*, also known as sensitivity, was defined as the ratio of the number of correct positive predictions to the total number of positives, reflecting the effectiveness of the model in predicting positive cases.
(7)$$Recall=\frac{TP}{TP+FN}$$

The *F*1 score was the reconciled mean of precision and recall, responding to the association of the predicted outcome with positive cases in the test set [60].
(8)$$F1\;score=\frac{2TP}{2TP+FP+FN}$$

With the default classification threshold, *p* > 0.50 is categorized as positive. However, this default threshold is often unsuitable for dealing with unbalanced data. To estimate the optimal threshold, we used the closest top-left method to select the point close to the upper-left corner of the ROC curve as the optimal threshold and reported the above one-dimensional performance metrics at the optimal threshold [61].

## 3. Results

### 3.1. Descriptive Results

Of the 3380 children under five years of age in this study, 1342 (39.70%) had stunted growth, with the mean age being 29.73 months. Most were boys (53.11%) and received breastfeeding for a duration of ≥6 months (Table 2). Regarding maternal characteristics, most (62.59%) mothers were not employed. Approximately half of the mothers (50.41%) had received a primary education, as had 45.40% of their partners. Household and community characteristics indicated that around 16.45% of the children came from the poorest households, almost half (46.19%) did not have access to improved water sources, the majority (76.36%) resided in rural areas, and about one-third (30.86%) hailed from the Southern Region (Table 2).

The prevalence of stunting was highest in the Highlands Region (58.97%) compared to other regions, and stunting prevalence among children from the poorest households (53.60%) was almost twice as high as that of children from the richest households (25.39%). Furthermore, the results of the χ2 test and Wilcoxon rank sum test showed that variables such as children’s birth order, early breastfeeding, occurrence of diarrhea and fever in the last two weeks, and the age and marital status of the mother and her partner were not significantly associated with child stunting, and thus, they were excluded from the follow-up study (Table 2).

The prevalence of stunting among children in PNG varied across provincial division, with the Southern Highlands showing the highest rates, while Manus and the National Capital District were less affected by stunting. The Highlands Region provinces, including Southern Highlands, Enga, Hela, Western Highlands, Jiwaka, Chimbu, and Eastern Highlands, also exhibited higher stunting rates compared to other provinces (Figure 2).

### 3.2. Feature Selection Results

Figure 3 presents the process of feature selection using the LASSO and RF-RFE methods. For LASSO, the model achieved the best AUC value (AUC: 0.669) at λ = 0.0051, resulting in the shrinkage of regression coefficients for 34 features to 0 and representing approximately 59.6% of all features. On the other hand, using RF-RFE, the model attained the best AUC value (AUC: 0.672) after removing the first 27 least important features, accounting for about 47.4% of all features.

### 3.3. Hyperparameter Tuning Results

Table 3 summarizes the best hyperparameters of CTree, SVM-RBF, and XGBoost models under 10-fold cross-validation using the GS or BO-GPR strategy. With the LASSO optimal feature subset, SVM-RBF demonstrated the best prediction performance in the training set (AUC: 0.671). Furthermore, the performance of CTree, SVM-RBF, and XGBoost in the training set improved after FS.

### 3.4. Evaluation of the Prediction Models

Table 4 and Figure 4 summarize the final performance of LR, CTree, SVM-RBF, and XGBoost using the test set. The results indicate that XGBoost, under the LASSO FS method, provided the best prediction performance (AUC: 0.765; 95% CI: 0.714–0.819), and the model’s accuracy, precision, recall, and F1 scores at the optimized threshold (0.487) were 0.728, 0.715, 0.628, and 0.669. Moreover, CTree exhibited the worst performance without using the FS method (AUC: 0.695; 95% CI: 0.639–0.750) (Table 4 and Figure 4).

The final performance of all models improved after using feature selection, indicating that the FS method effectively eliminated noise or redundant information while preserving crucial features of the original model (59.6% dimensionality reduction for LASSO and 47.4% dimensionality reduction for RF-RFE). The impact of feature selection varied depending on the optimized model: for CTree and XGBoost, the performance was best with LASSO, while for LR and SVM-RBF, the performance was optimized using RF-RFE (see Table 4 and Figure 4).

### 3.5. Model Interpretation

SHapley Additive exPlanations (SHAP) is a feature attribution method based on a game-theoretic framework that helps reveal the decision-making process of complex “black-box models” such as XGBoost. As mentioned above, we used the SHAP value method to explain the XGBoost prediction model under the LASSO optimal feature subset.

#### 3.5.1. SHAP Summary Plots

The SHAP summary chart sorted the characteristics vertically from highest to lowest based on the mean absolute SHAP values. We selected the top 15 characteristics to illustrate their relative importance in predicting stunting in children (refer to Figure 5). Notably, living in the Highlands Region, the child’s age, belonging to the wealthiest family, and having a larger or smaller birth size were identified as the top five most significant factors.

Additionally, the SHAP summary chart represents each child’s features as points, which are colored according to their feature values, ranging from low (blue) to high (red). For binary feature vectors, red dots indicated the presence of the corresponding feature in the individual child. The SHAP value on the horizontal axis reflects the contribution of the feature to the model output. Higher SHAP values indicate a greater likelihood of stunting. Specifically, children from the Highlands Region, those with smaller birth sizes, or those in the poorest households had SHAP values > 0 for the corresponding characteristics, indicating a higher probability of stunting. In contrast, children from the wealthiest families who were female or of larger birth size had SHAP values < 0 for the corresponding trait, indicating a lower probability of stunting (see Figure 5).

#### 3.5.2. SHAP Dependence Plot of Child’s Age

To provide a more intuitive view of the relationship between feature values and the model’s expected output, we constructed a dependency plot for child age (a continuous variable) versus SHAP values. The plot included points representing different individual children. The smoothed line of partial regression demonstrated a positive association between child age and SHAP values when children were ≤24 months old. After 24 months of age, SHAP values stabilized and remained positive for the vast majority of children (Figure 6).

## 4. Discussion

Based on the nationally representative PNG DHS 2016–2018 dataset, our study suggests that the LASSO-XGBoost combination had the best performance in predicting stunting among children under five years old in PNG (AUC: 0.767, 95% CI: 0.714–0.819). The optimal model identified living in the Highlands Region, the age of the child, being in the wealthiest household, and having a larger or smaller birth size as the top five most important characteristics for predicting stunting in children, reflecting the complexity of the causes of stunting. Critical findings of the study include the following.

Firstly, the study found that children residing in the Highlands Region were at a very high risk of stunting, and stunting was also most prevalent in the region (58.97%), which is similar to the findings of an earlier study using the 2009–2010 PNG HIES [19]. Food insecurity is one of the key underlying causes of child malnutrition [34]. In the Highlands Region, children are extremely vulnerable to food insecurity caused by events such as extreme weather and social conflict [62,63]. Long-term food deprivation [64] is associated closely with chronic malnutrition. Diets in the Highlands excessively rely on mono-foods (such as starchy foods like sweet potatoes and sago, among others) [63,65], and nutritionally unbalanced feeding practices could also contribute to linear growth deficits in children [14,64].

Secondly, the study highlights the strong association between household wealth, child age, and stunting. Children from wealthier households face a lower risk of stunting, which is potentially due to their better resilience against food insecurity [66], improved access to healthcare facilities [67], and ability to access high-protein foods [68,69]. Furthermore, the study observed a rapid increase in the risk of stunting in children aged 0–24 months, which is in line with previous cross-country studies [3]. This emphasized the urgency of early intervention to prevent stunting from exacerbating the cycle of deprivation, especially among the most vulnerable groups of children living in poverty [70].

Finally, the study underscores the significance of birth size in determining a child’s growth potential. A smaller birth size is associated with a higher risk of stunting in children, while a larger birth size is a protective factor. Maternal malnutrition during pregnancy could be a potential cause of smaller birth sizes, leading to altered fetal and placental growth patterns and contributing to impaired fetal growth [71,72]. Existing studies have emphasized the importance of intrauterine health in preventing stunting in children [73]. Therefore, it is necessary to focus on and improve the nutritional status of pregnant women and women of childbearing age (15–49 years) in PNG. However, it is important to emphasize that exploring the relationship between birth size and stunting still requires caution due to the subjective assessment of the child’s birth size by mothers.

Moreover, being female, mothers’ exposure to mass media, mothers’ secondary education level, and their partners’ higher education level were discovered to be protective factors against stunting. Evidence from the Highlands Margins of PNG suggests that gender heterogeneity in stunting may be attributed to girls’ growth strategies, which prioritize growth over maintenance to meet future reproductive potential [74]. Meanwhile, mothers and their partners with high levels of education were likely to have better incomes, leading to improved nutrition for their children [75]. And mothers exposed to mass media were more likely to acquire knowledge about proper modern healthcare practices and correct inappropriate attitudes [76].

In conclusion, our results show that the combination of ML and FS techniques provides a better classification of stunting. After BO-BPR hyperparameter tuning with 10-fold cross-validation, the LASSO-XGBoost model achieved the best predictive performance compared to traditional logistic regression (LR) (AUC: 0.765; 95% CI: 0.714–0.819). Particularly, LASSO and RF-RFE facilitated efficient ML learning by removing redundant, noisy information, resulting in a substantial dimensionality reduction of 59.6% and 47.4%, respectively. Thus, we suggest prioritizing the best combination of LASSO and XGBoost when stunting in PNG children is a central concern for prediction.

This study had several important strengths. Firstly, the data were derived from the nationally representative PNG DHS 2016–2018. Secondly, the study used ML algorithms and FS techniques to make better predictions, which have not been widely used in related research in PNG and could provide lessons for researchers conducting research on similar topics in PNG and other Pacific Island countries. Nevertheless, some potential limitations remain. Firstly, the SHAP value method employed in the study provided correlation analysis but could not establish causal inferences; therefore, the interpretability of the results is still limited. Secondly, although we tried to include as comprehensive a set of variables as possible, we could not exclude residual confounding caused by unmeasured variables such as the mother’s height and weight. Moreover, some of the children’s information was derived from their mother’s recollections (for instance, the occurrence of diarrhea and fever in the child in the last two weeks), and there may be a recollection bias.

## 5. Conclusions

Based on cross-sectional data from the nationally representative PNG DHS 2016–2018, this study used the ML algorithm with FS techniques to identify the optimal model and crucial factors for predicting stunting in children under five years of age in PNG. The results show that the combination of LASSO and XGBoost had the best predictive performance. Living in the Highlands Region, the child’s age, being in the richest household, and having a larger or smaller birth size emerged as the top five important characteristics for predicting stunting. The findings emphasize the importance of early-life interventions to prevent stunting, especially for the most vulnerable groups of children in the marginalized Highlands Region. Therefore, there is an imperative for more robust public health policies and interventions aimed at enhancing maternal nutrition and disseminating accurate knowledge of modern healthcare practices to promote maternal and child health and well-being in PNG.

## Figures and Tables

**Figure 1 children-10-01638-f001:**
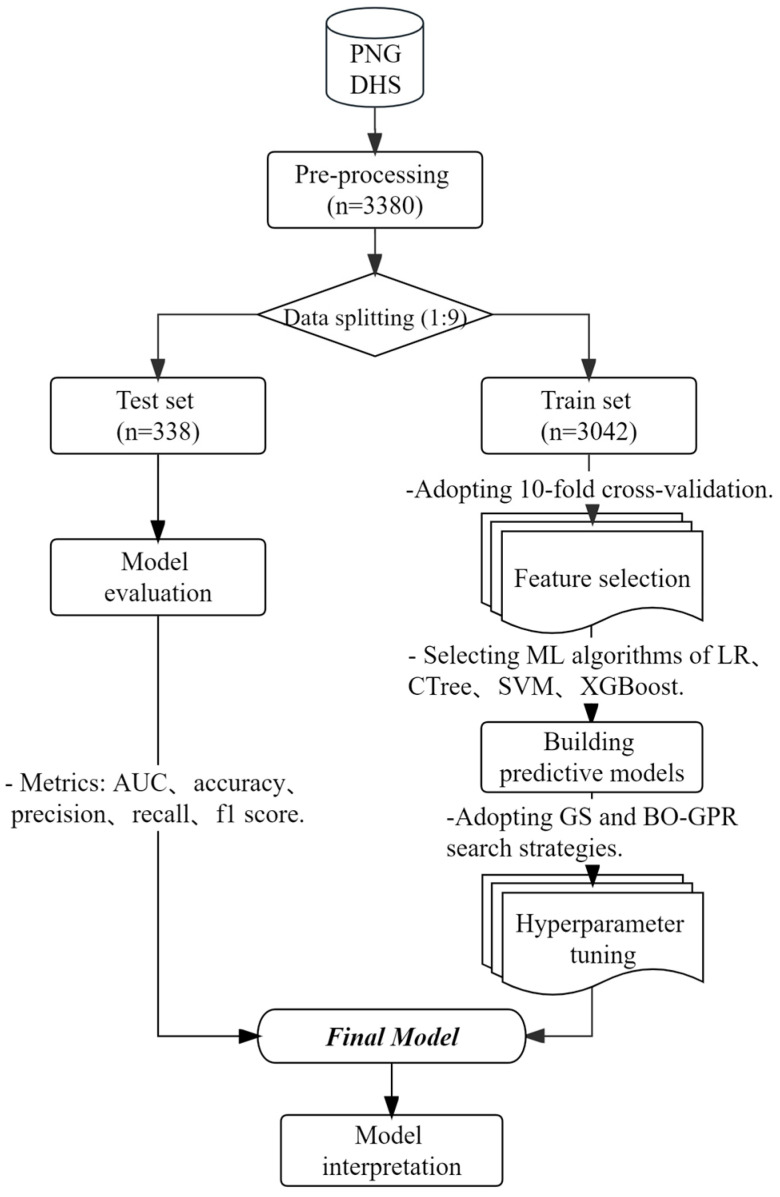
Analysis flow diagram.

**Figure 2 children-10-01638-f002:**
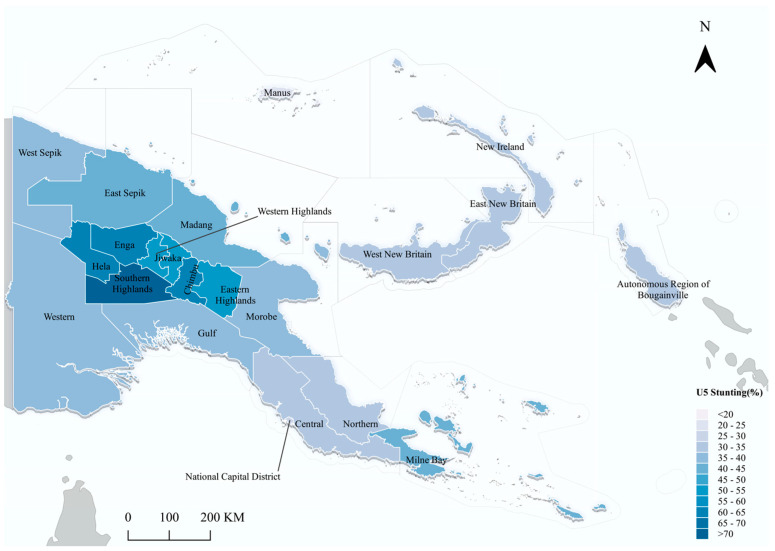
Spatial distribution of stunting rates for children under five years of age by provincial-level divisions in Papua New Guinea; 2016–2018 PNG DHS.

**Figure 3 children-10-01638-f003:**
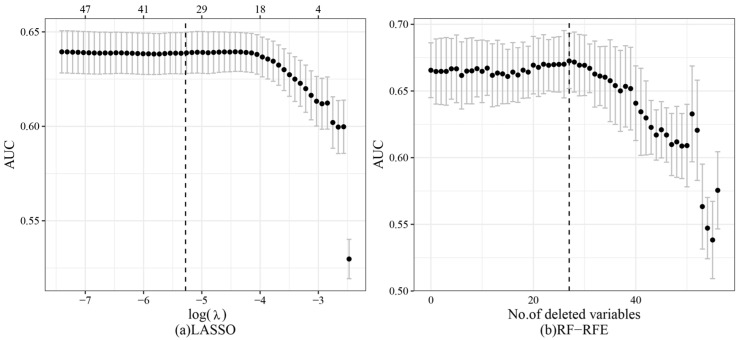
Feature selection process for LASSO and RF-RFE. (**a**) feature selection process for LASSO; (**b**) feature selection process for RF-RFE.

**Figure 4 children-10-01638-f004:**
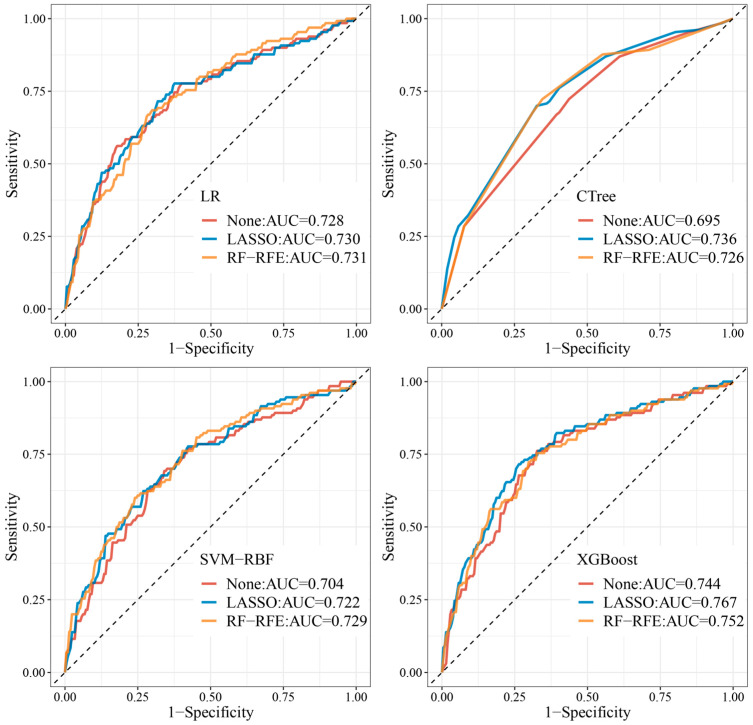
ROC curve and AUC performance of the prediction models using the test set.

**Figure 5 children-10-01638-f005:**
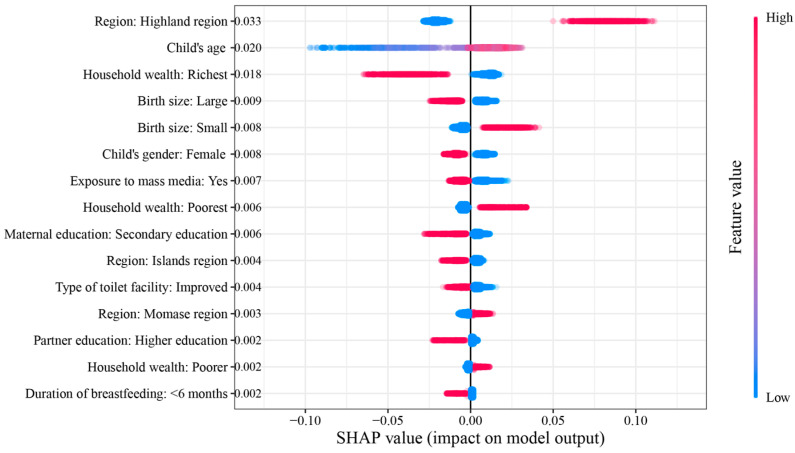
SHAP summary plot with top 15 contributing features for XGBoost models.

**Figure 6 children-10-01638-f006:**
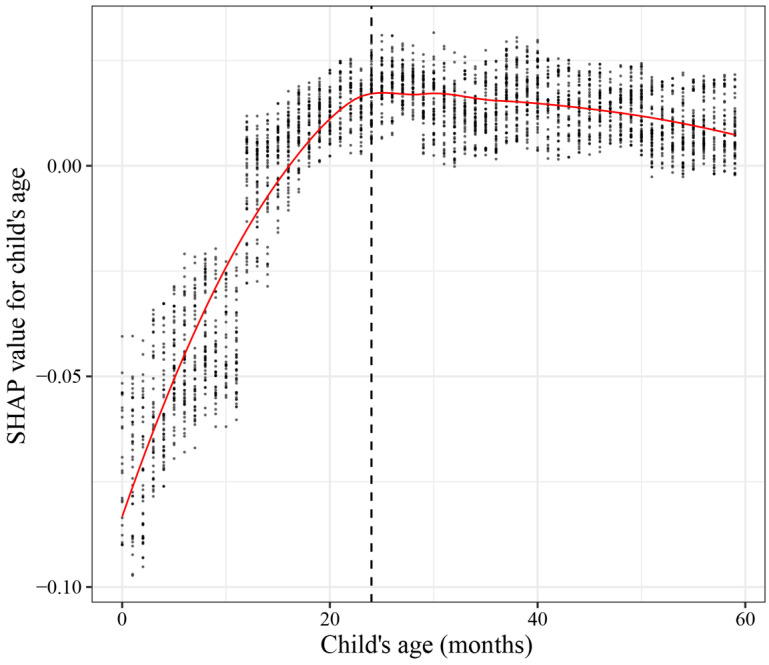
SHAP dependence plot of child’s age.

**Table 1 children-10-01638-t001:** Hyperparameter tuning range for XGBoost.

Hyperparameter	Range	Type
Eta	(0.01, 0.3)	Real
Gamma	(0, 0.2)	Real
Subsample	(0.1, 1)	Real
Colsample bytree	(0.1, 1)	Real
Nrounds	[1, 200]	Integer
Maxdepth	[1, 20]	Integer
Min child weight	[1, 20]	Integer

**Table 2 children-10-01638-t002:** Prevalence of stunting in children under 5 in Papua New Guinea by characteristics; PNG DHS 2016–2018.

			Stunted		
Variables	N	Frequency (%)/Mean (SD)	No (%)	Yes (%)	*p*-Values
Individual characteristics					
Child’s age (months)	3380	29.73			<0.001
Child’s gender					<0.01
Male	1795	53.11	57.83	42.17	
Female	1585	46.89	63.09	36.91	
Birth size					<0.001
Average	1215	38.36	65.93	34.07	
Large	1337	42.22	60.13	39.87	
Small	615	19.42	51.22	48.78	
Birth order	3380	3.15			0.069
Duration of breastfeeding					<0.001
Never breastfed	168	7.19	60.71	39.29	
<6 months	367	15.71	76.02	23.98	
≥6 months	1801	77.10	59.74	40.26	
Early breastfeeding					0.280
No	919	41.81	63.87	36.13	
Yes	1279	58.19	61.61	38.39	
Had diarrhea in the past 2 weeks					0.925
No	2683	84.74	60.27	39.73	
Yes	483	15.26	60.04	39.96	
Had fever in the past 2 weeks					0.867
No	2489	78.72	60.39	39.61	
Yes	673	21.28	60.03	39.97	
Maternal characteristics					
Maternal age (years)	3380	30.16			0.848
Partner’s age (years)	2961	35.03			0.547
Maternal employment status					<0.001
Not employed	2101	62.59	57.45	42.55	
Employed	1256	37.41	64.81	35.19	
Partner’s employment status					<0.001
Not employed	1328	43.60	55.72	44.28	
Employed	1718	56.40	63.50	36.50	
Maternal occupation					<0.001
No occupation	2124	63.46	57.63	42.37	
Professional/technical/managerial	161	4.81	78.88	21.12	
Clerical	66	1.97	69.70	30.30	
Sales	161	4.81	72.67	27.33	
Agricultural	560	16.73	56.43	43.57	
Services	257	7.68	68.87	31.13	
Skilled manual	7	0.21	85.71	14.29	
Unskilled manual	11	0.33	54.55	45.45	
Maternal marital status					0.456
Never Married/divorced/separated	274	8.11	62.41	37.59	
Married/living together	3106	91.89	60.11	39.89	
Maternal religion					0.843
Non-Christian/no religion	29	0.86	58.62	41.38	
Christian	3343	99.14	60.42	39.58	
Maternal education level					<0.001
No education	647	19.14	48.53	51.47	
Primary education	1704	50.41	59.10	40.90	
Secondary education	918	27.16	69.17	30.83	
Higher education	111	3.28	73.87	26.13	
Partner’s education level					
No education	458	15.17	46.72	53.28	
Primary education	1371	45.40	58.35	41.65	
Secondary education	953	31.56	64.85	35.15	
Higher education	238	7.88	76.05	23.95	
Exposure to mass media					<0.001
No	1646	49.03	53.95	46.05	
Yes	1711	50.97	66.69	33.31	
Maternal age of first birth (years)	3380	21.17			<0.01
Household characteristics					
Sex of househead					0.061
Male	2892	85.56	59.65	40.35	
Female	488	14.44	64.14	35.86	
Household wealth					<0.001
Poorest	556	16.45	46.40	53.60	
Poorer	531	15.71	49.91	50.09	
Middle	653	19.32	60.49	39.51	
Richer	809	23.93	61.80	38.20	
Richest	831	24.59	74.61	25.39	
Number of under-5 children	3380	3.35			<0.05
Number of household members	3380	6.93			<0.05
Type of toilet facility					<0.001
No facility	683	20.47	61.35	38.65	
Unimproved	1579	47.33	54.40	45.60	
Improved	1074	32.19	68.53	31.47	
Source of drinking water					<0.001
Unimproved	1558	46.18	54.36	45.64	
Improved	1816	53.82	65.42	34.58	
Type of cooking fuels					<0.001
Polluting fuels	3058	91.64	58.70	41.30	
Clean fuels	279	8.36	78.85	21.15	
Distance to health facility					<0.001
Not a big problem	1509	45.14	64.88	35.12	
Big problem	1834	54.86	56.32	43.68	
Community characteristics					
Region					<0.001
Southern Region	663	19.62	65.68	34.32	
Highland Region	1043	30.86	41.03	58.97	
Momase Region	799	23.64	59.32	40.68	
Islands Region	875	25.89	69.37	30.63	
Area					<0.001
Rural	2581	76.36	56.95	43.05	
Urban	799	23.64	71.09	28.91	

**Table 3 children-10-01638-t003:** Optimal value of each hyperparameter searched by the optimization strategy.

	Trainset (Cross-Validation)	
Models	Optimal Hyperparameters	AUC
**None**		
CTree	maxdepth = 5, mincriterion = 0.950	0.639
XGBoost	nrounds = 12, eta = 0.153, gamma = 0.091, subsample = 0.807, colsample bytree = 0.995, maxdepth = 6, min child weight = 5	0.644
SVM-RBF	*C* = 2^−5^, *σ* = 2^−15^	0.658
**LASSO**		
CTree	maxdepth = 7, mincriterion = 0.900	0.642
XGBoost	nrounds = 12, eta = 0.012, gamma = 0.199, subsample = 0.694, colsample bytree = 0.811, maxdepth = 7, min child weight = 13	0.653
SVM-RBF	*C* = 2^15^, *σ* = 2^−15^	0.671
**RF-RFE**		
CTree	maxdepth = 4, mincriterion = 0.990	0.646
XGBoost	nrounds = 19, eta = 0.149, gamma = 0.058, subsample = 0.909, colsample bytree = 1, maxdepth = 20, min child weight = 18	0.666
SVM-RBF	*C* = 2^−1^, *σ* = 2^−5^	0.666

**Table 4 children-10-01638-t004:** Performance summary of the prediction models.

Models	Test Set					
Metric	AUC (95% CI)	*Accuracy*	*Precision*	*Recall*	*F*1 *Score*	Threshold
**None**						
LR	0.728 (0.672–0.785)	0.675	0.731	0.559	0.633	0.370
CTree	0.695 (0.639–0.750)	0.630	0.669	0.515	0.582	0.426
XGBoost	0.744 (0.690–0.798)	0.707	0.762	0.593	0.667	0.400
SVM-RBF	0.704 (0.646–0.761)	0.672	0.692	0.559	0.619	0.363
**LASSO**						
LR	0.730 (0.674–0.787)	0.692	0.708	0.582	0.639	0.391
CTree	0.736 (0.682–0.789)	0.683	0.700	0.572	0.630	0.459
XGBoost	0.767 (0.714–0.819)	0.728	0.715	0.628	0.669	0.487
SVM-RBF	0.722 (0.666–0.778)	0.672	0.677	0.561	0.613	0.346
**RF-RFE**						
LR	0.731 (0.676–0.785)	0.695	0.685	0.589	0.633	0.394
CTree	0.726 (0.672–0.781)	0.681	0.723	0.566	0.635	0.343
XGBoost	0.752 (0.698–0.806)	0.710	0.723	0.603	0.657	0.388
SVM-RBF	0.729 (0.674–0.785)	0.692	0.615	0.597	0.606	0.367

## Data Availability

Access to the dataset is available at https://dhsprogram.com/data/dataset/Papua-New-Guinea_Standard-DHS_2017.cfm?flag=0 (accessed on 17 July 2023).

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
