# Peer review of "Machine Learning Algorithms for Predicting Stunting among Under-Five Children in Papua New Guinea"

_children, 2023, doi:10.3390/children10101638_

Round 1
Reviewer 1 Report
Typo "...2016-18 Papua New..." Line85;
What is the urgency of this research? it should be emphasized at the end of the introduction.
What is the impact of this study on the wider community? besides the people of PNG. It should appear as an advantage of this research and place it at the end of the discussion.
Why does Authors only analyze for 2016-2018 data? why not until 2022?
Permission to use data and research ethics must be declared in more detail and further.
Extensive editing of English language required and certificate of English correction needed.
Author Response
Please see the PDF attachment.

Reviewer 2 Report
This is a very well conducted study and well presented manuscript. I do have only some minor comments:
- Line 113: Please use the DHS hyperlink and not a Google link.
- Table 2: Please correct into "birth size".
- Lines 396-400: Please elaborate more on the limitations of this study.
Author Response
Please see the PDF attachment.

Reviewer 3 Report
This is an well-written, interesting study with high novelty. Some points slhould be addressed:
- The reported interviews were face-to-face with qualified personel? And what personel?
- Anthropometric measurements should be accompanied with relevant references.
- The inclusion and exclusion criteria of study enrolment should be emphasized.
- The normality distribution test should be reported.
- All Figures should be improved concerning their resolution and size.
- Moderate English language editing is required.
-
Moderate English language editing is required.
Author Response
Please see the PDF attachment.

Round 2
Reviewer 1 Report
Authors revised the manuscript and consider to accept.